# Replacing the Addition of Sulfite in Mustard Pickle Products by High-Hydrostatic-Pressure Processing to Delay Quality Deterioration during Storage

**DOI:** 10.3390/foods12020317

**Published:** 2023-01-09

**Authors:** Hung-I Chien, Yi-Chen Lee, Yu-Fan Yen, Pi-Chen Wei, Chiu-Chu Hwang, Chia-Hung Kuo, Feng-Lin Yen, Yung-Hsiang Tsai

**Affiliations:** 1Department of Seafood Science, National Kaohsiung University of Science and Technology, Kaohsiung 811213, Taiwan; 2Department of Bioscience and Biotechnology, National Taiwan Ocean University, Keelung 202301, Taiwan; 3Department of Fragrance and Cosmetic Science, Kaohsiung Medical University, Kaohsiung 807378, Taiwan

**Keywords:** high-hydrostatic-pressure, mustard pickle, quality, lactic acid bacteria, high-throughput sequencing

## Abstract

This study aimed to assess the use of the high-hydrostatic-pressure (HHP) method (200–600 MPa, 5 min) for bleaching mustard pickle products as an alternative to the conventional method of sulfite addition. The aerobic plate count (APC) and lactic acid bacteria count (LAB) of the samples decreased with the increase in pressure, and the yeast count decreased to no detectable levels. Next, compared with the control group (no high-pressure treatment) the *L** (lightness), *W* (whiteness), Δ*E* (color difference), and texture (hardness and chewiness) of the HHP-processed samples, which increased significantly with increasing pressure, while the *a** (redness) and *b** (yellowness) values decreased slightly. This indicates that HHP processing gave the mustard pickle a harder texture and a brighter white color and appearance. Furthermore, when the mustard pickle was treated with HHP 400 and 600 MPa for 5 min and stored at 25 °C for 60 days, it was found that the APC and LAB counts in the HHP-processed group recovered rapidly and did not differ from those in the control group (the non-HHP treated group) but significantly delayed the growth of yeast, the increase in pH value, and total volatile basic nitrogen (TVBN). The high-throughput sequencing (HTS) analysis revealed that the predominant bacterial genera in the non-HHP-treated mustard pickle were *Lactiplantibacillus* (74%), *Lactilactobacillus* (12%), and *Levilactobacillus* (6%); after 60 days of storage, *Companilactobacillus* (80%) became dominant. However, after 60 days of storage, *Lactiplantibacillus* (92%) became dominant in the samples processed at 400 MPa, while *Levilactobacillus* (52%), *Pediococcus* (17%), and *Lactiplantibacillus* (17%) became dominant in the samples processed at 600 MPa. This indicated that the HHP treatment changed the lactic acid bacterial flora of the mustard pickle during the storage period. Overall, it is recommended to treat the mustard pickle with HHP above 400 MPa for 5 min to improve its texture and color and delay the deterioration of quality during storage. Therefore, HHP technology has the potential to be developed as a treatment technique to replace the addition of sulfite.

## 1. Introduction

Mustard pickle is the most common fermented vegetable in Taiwan. The main manufacturing process includes placing the mustard pickle in a vat, then sprinkling a layer of salt equivalent to approximately 14% of the weight of the mustard, then repeating the process of layering the mustard and salt, and finally pressing the top mustard layer with a stone. The fermentation duration lasts about 3–6 months. After fermentation, the outermost leaves of the mustard are removed and the stems are soaked in sulfite solution for bleaching, before being finally drained to get the finished mustard pickle product [1,2]. Subsequently, the mustard pickle products are transported to traditional markets where they are normally sold at room temperature [2]. In general, the average salt content and water activity of mustard pickle products sold in retail market are about 4.9% and 0.944, respectively, which is not enough to achieve a completely preservative effect [2]. In addition, the main purpose of adding sulfites to mustard pickle products is bleaching, and the secondary purpose is an antibacterial effect [2].

High-hydrostatic-pressure (HHP) processing refers to packaging food products in a soft, sealed container with a liquid (water) as the medium for transferring the pressure in order to process the food with physical treatment in a high-pressure environment. When food is subjected to pressure > 300 MPa, the microorganisms in the food will be destroyed and die [3]. Therefore, HHP processing serves to reduce the counts of pathogens and spoilage bacteria in food to extend the storage life while maintaining the original nutrients and natural flavor [4]. In general, the cell wall of Gram-positive bacteria is thicker than that of Gram-negative bacteria, and thus the resistance to high pressure is higher. Additionally, spores produced by Gram-positive bacteria are the most resistant and some spores can even survive pressures of 1000 MPa [3,4]. Currently, HHP treatment has been widely adopted in the preservation of vegetables and fruit juices, meat products, and shellfish processing.

Studies have been conducted to examine the impacts of HHP processing on the quality of fermented vegetables. For example, Sohn and Lee [5] treated kimchi with HHP processing and found that 400 MPa HHP had no effect on the texture of kimchi while 600 MPa increased the cutting force of kimchi. HHP-processed kimchi (400 MPa for 10 min) maintained 10^3^ CFU/mL of viable bacteria when stored at 20 °C for four weeks, and HHP above 400 MPa prevented over-acidification and bulging induced by CO_2_ build-up during extended storage. Therefore, HHP processing can delay the overripening of kimchi during transportation and storage. Park et al. [6] suggested that HPP processing of kimchi with at least 500 MPa for 5 min was required to completely inactivate norovirus; however, during this process, the counts of viable and lactic acid bacteria only decreased by 1 log value. Next, there was no significant difference in redness, yellowness, and hardness between HHP-processed and unprocessed kimchi (100–500 MPa), while HPP-processed kimchi had a good sensory evaluation. Therefore, HHP processing at 500 MPa for 5 min can be employed as a processing method for commercial kimchi to inactivate norovirus without affecting the quality [6]. Li et al. [7] contend that HHP processing of sour Chinese cabbage (SCC) at 200 MPa has no effect on microorganisms, while HHP processing at 400 and 600 MPa obviously reduces the number of viable bacteria, lactic acid bacteria, and yeast. Moreover, after the 600 MPa-processed SCC was stored for 90 days (at 4 °C, 27 °C, and 37 °C), the counts of viable bacteria decreased to undetectable levels, and the counts of yeast in the 400 and 600 MPa treatment groups decreased to undetectable levels after two days of storage. Therefore, HHP processing at 600 MPa can be considered an alternative procedure for SCC processing and preservation. Peñas et al. [8] observed a decrease of about 4–5 log values in the counts of viable and lactic acid bacteria after applying HHP processing at 300 MPa to sauerkraut for 10 min; the log values of the counts increased gradually after three months of refrigeration; however, they remained lower than the unprocessed group of the same storage duration. Therefore, HHP can be regarded an effective procedure to improve the microorganism quality and storage life of sauerkraut [8]. In summary, it is revealed that the impacts of HHP processing on the microorganisms and physical properties of different types of fermented vegetables varied considerably.

As many microorganisms do not grow in culture media, the findings from conventional media culture methods cannot elucidate the actual microorganism bacterial flora in foods [9,10]. Moreover, multiple molecular biotechnologies, such as real-time polymerase chain reaction (real-time PCR), denaturing gradient gel electrophoresis (DGGE), and terminal restriction fragment length polymorphism (TRFLP), cannot fully detect the species and proportion of microorganisms in a specific food. However, the adoption of high-throughput sequencing (HTS) techniques can effectively and accurately analyze the microbiota in foods [11]. Proposals for the PacBio single-molecule real-time (SMRT) sequencing platform, which is powered by third-generation high-throughput sequencing technology, were recently presented, and they utilized various sequencing methods to generate long-read sequences (10–15 kb on average) of base pair data. The read lengths range between 3000 and 15,000 bp with 99% accuracy [12]. This suggests that the SMRT sequencing technology is an excellent tool for generating 16S rRNA gene sequence data to identify bacterial diversity and community structure at the species level.

Previous studies have indicated that the sulfite content of mustard pickle products sold in Taiwan is above 100 ppm, which significantly exceeds the limit of 30 ppm for food additives and may cause asthma and other allergic reactions in humans [2,13]. Therefore, the excess residual sulfite in mustard pickle products is a dangerous allergen causing health hazards to consumers. In addition, only a very few studies have been conducted to apply HHP technology to the manufacturing of mustard pickle products. Therefore, this study was to assess the use of the HHP method (a non-thermal processing physical pasteurization technique) for bleaching and preserving mustard pickle products as an alternative to the conventional method of sulfite addition.

## 2. Materials and Methods

### 2.1. Samples

Mustard pickle (*Brassica juncea*) products (5 kg in total) were purchased from a mustard pickle manufacturing plant (GPS: N23°38.383′ E120°27.993′) that was fermented for four months without sulfite addition. The samples were wrapped in crushed ice and transported to our research lab. After preliminary analysis, the pH value, water activity, water content, salt content, and titratable acidity of the sample were 4.84, 0.956, 92.36%, 2.68%, and 0.35%, respectively. We then cut the samples into 100 g pieces and placed them in vacuum bags (nylon/linear low-density polyethylene) for subsequent high-pressure treatment.

### 2.2. HHP Technology

The vacuum-packed samples were put in a hydrostatic apparatus (HPP-6.2L, Baotou Kefa Co., Ltd., Baotou, Inner Mongolia, China) with a capacity of 6.2 L for HPP processing. Deionized water was applied to the fluid for transferring the pressure. The pressurization rate was around 150 MPa/min, and the decompression time was about 9–14 s. The processing duration did not contain the time for pressure build-up and release. HHP treatment conditions included 200, 300, 400, 500, and 600 MPa for 5 min. The control (non-HHP treated) group comprised samples that were vacuum-packed but not processed at high pressure. After the high-pressure processing, we then analyzed the change in aerobic plate count (APC), lactic acid bacteria count (LAB), yeast, color, and texture of the mustard pickle samples.

### 2.3. Storage Test

The samples pressurized at 400 and 600 MPa were stored at 25 °C for 60 days and sampled every six days to analyze the changes in APC, LAB, yeast, pH, and total volatile basic nitrogen (TVBN). The aim of storing at 25 °C was to simulate the way mustard pickle products are stored and sold at room temperature in the conventional Taiwanese market.

### 2.4. Microbiological Analysis

The APC assay was performed using a smear culture method to count live natural microorganisms. A sample of the chopped mustard pickle (10.0 g) was homogenized in a homogenizer bottle that contained 90.0 mL of sterile 0.85% NaCl solution. Then, the homogenate was diluted 10 times in a row with sterile 0.85% NaCl solution, and 0.1 mL of the diluted solution was spread on trypticase soy agar (TSA) (Difco, BD, Sparks, MD, USA) plates in duplicate. After incubation at 35 °C for 48 ± 2 h, the colonies were counted. Further, 0.1 mL of the APC dilution was spread on de Man, Rogosa, and Sharpe agar (MRSA) (Difco, BD, Sparks, MD, USA) for incubation at 30 °C for 48 ± 2 h, and the colonies were counted. Yeast was analyzed using a 3M Petrifilm Yeast and Mold Count Plate (3M Microbiology, St. Paul, MN, USA) in accordance with the guideline manual.

### 2.5. Appearance and Color Value

The mustard pickle was cut into appropriate sizes and placed in a plastic Petri dish with a white background. Its appearance was recorded by taking pictures with a cell phone (iPhone 8; Apple Inc., Cupertino, CA, USA). To analyze the color values, the mustard pickle was cut into 2 × 2 cm squares and assessed in reflection mode at 25 ± 1 °C using a colorimeter (NR-12A, NIPPON DENSHOKU Industries Co., LTD, Tokyo, Japan). A white reference plate and a D65 light source were used, with a 0° viewing angle. Regarding the obtained data, the *L** value represents lightness, the *a** value denotes redness ranging from negative (green) to positive (red), and the *b** value represents yellowness ranging from negative (blue) to positive (yellow). In addition, the whiteness (*W*) and the total color difference (Δ*E*) were calculated using the following equations, where *L_0_**, *a_0_**, and *b_0_** denote the data of control samples:(1)$$W=100-\sqrt{{\left(100\;-\;L*\right)}^{2}+a{*}^{2}+\;b{*}^{2}}$$
(2)$$\Delta E=\sqrt{{\left(L*-\;L_{0}*\right)}^{2}+{\left(a*-\;a_{0}*\right)}^{2}+(b*-\;b_{0}*{)}^{2}}$$

### 2.6. Texture

We made some modifications in accordance with the method proposed by Lee et al. [14] and performed texture profile analysis (TPA) using a Brookfield CT3 texture analyzer (AMETEK Brookfield, Middleborough, MA, USA) to determine the hardness, cohesiveness, springiness, and chewiness. The samples that were cut into 2.0 × 2.0 cm squares were placed on the platform and measured using a cylindrical probe (TA44, 4 mm diameter) with a prediction speed, test speed, and return speed of 1 mm/s and a compression distance of 3 mm.

### 2.7. pH and TVBN Values

A total of 5 g of sample was homogenized with 25 mL of deionized water by a homogenizer (Polytron PT 3000, East Lyme, CT, USA) for 2 min, and the pH value of the filtrate sample was then determined with a HORIBA pH meter F-71S (Kyoto, Japan). The TVBN contents in the mustard pickle samples were extracted using trichloroacetic acid in accordance with the method proposed by Chen et al. [15], and TVBN values were then determined with Conway’s dish method proposed by Cobb et al. [16].

### 2.8. HTS Analysis

Finely crushed mustard pickle samples (100 g) were weighed, and the juice was squeezed out by covering with gauze under sterile conditions. The liquid was centrifuged (4000× *g*, 10 °C, 10 min), and the supernatant was discarded. After mixing the precipitate with 9 mL of phosphate-buffered saline, the centrifugation step was repeated, and the supernatant was discarded. Bacterial genomic DNA was extracted from the precipitate using a QIAamp PowerFecal DNA Kit (QP; Qiagen, Hilden, Germany) according to the manufacturer’s instructions.

To the extracted genomic DNA, we added a set of 16-nucleotide barcodes. Forward (5′-AGRGTTYGATYMTGGCTCAG-3′) and reverse (5′-RGYTACCTTGTTACGACTT-3′) primers designed specifically for SMRT sequencing of the full-length 16S rRNA gene were used to perform PCR (2720 Thermal Cycler, Applied Biosystems, Foster City, CA, USA) to amplify the bacterial 16S rRNA gene. The PCR amplification conditions were as follows: initial denaturation at 95 °C for 3 min, main denaturation step at 95 °C for 30 s, annealing at 55 °C for 30 s, and extension at 72 °C for 60 s, for 27 cycles. The final step involved heating at 72 °C for 5 min. The quality of the amplified products was examined using 1% agarose gel electrophoresis and spectrophotometry (Nanodrop 1000, Thermo Fisher Scientific, Waltham, MA, USA).

The amplified products were sequenced on a PacBio RS II (Pacific Biosciences, Menlo Park, CA, USA) with P6-C4 chemistry. Raw sequencing data were analyzed and refined using the quality clinical flow chart provided by Quantitative Insights into Microbial Ecology version 1.7 to ensure high accuracy in detecting the operational taxonomic units (OTUs). Representative sequences were identified by aligning the high-quality sequences extracted showing 100% clustering of sequence identity. OTUs with 97% similarity were selected using UCLUST software, and the representative sequences were submitted to the RDP classifier to obtain classifications at the phylum, class, order, family, and genus levels.

### 2.9. Statistical Analysis

Microorganisms, color, texture, pH, and TVBN values of mustard pickle samples were measured after immediate high-pressure treatment or during storage to analyze the differences between different pressure groups. All values were derived as the average ± standard deviation of three samples, and analysis of significant differences was conducted with the Duncan test and one-way ANOVA using statistical software SPSS version 22.0 (Armonk, NY, USA). A value of *p* < 0.05 presented a significant difference.

## 3. Results and Discussion

### 3.1. Effects of HHP Processing on the Microbial Quality of Mustard Pickle

The variations of APC, LAB, and yeast in the mustard pickle samples after HHP processing (200–600 MPa for 5 min) are presented in Table 1. It was found that, compared with the control group (6.99 log CFU/g), the APC decreased to 6.73, 6.32, 4.23, 4.03, and 3.81 log CFU/g after processing at 200, 300, 400, 500, and 600 MPa, respectively (namely, a reduction of 0.26, 0.67, 2.76, 2.96, and 3.18 log CFU/g, respectively), indicating a significant decrease in APC with increasing pressure (*p* < 0.05). The findings are similar to those reported by Sohn and Lee [5], namely, that the APC of kimchi decreased obviously with the increasing pressure, with a reduction of about 4 log values after processing at 600 MPa for 10 min. Li et al. [1] pointed out that processing SCC at 200 MPa had no effect on APC, while processing SCC at 400 and 600 MPa (for 10 min) obviously reduced APC by about 2.6 and 4.0 log values, respectively, which were similar to the results of this study. However, Park et al. [6] pointed out that while the APC of kimchi decreased with higher pressure, the count of organisms pressurized at a maximum pressure of 500 MPa (for 5 min) decreased by a mere 1 log value.

In terms of LAB variation, it was found that, compared with the control group (6.99 log CFU/g), the count of LAB bacteria decreased to 6.56, 6.53, 4.67, 3.97, and 3.75 log CFU/g (namely, a reduction of 0.43, 0.46, 2.32, 3.02, and 3.24 log CFU/g, respectively) after processing at 200, 300, 400, 500, and 600 MPa, respectively. Mustard pickles processed at 200 and 300 MPa (*p* > 0.05) exhibited no statistically significant differences in LAB counts. However, the LAB counts decreased obviously (*p* < 0.05) with the increasing pressure in mustard pickles processed at all higher pressures. The findings of this study are similar to those revealed by Sohn and Lee [5], namely, the LAB count of kimchi decreased significantly with higher pressure, with a reduction of about 5 log values after pressurizing at 600 MPa for 10 min. In addition, the LAB values for SCC were unchanged after treatment at 200 MPa, whereas the values significantly reduced by about 2.5 and 7.0 log values after processing at 400 and 600 MPa (10 min), respectively [7].

In terms of yeast variation, 1.41 log CFU/g of yeast was detected in the control group, while no yeast was detected in HHP-processed groups (200–600 MPa) (namely, a reduction of 1.41 log CFU/g). Our findings were similar to those revealed by Li et al. [7], namely, the higher the pressure and the duration of pressurization, a higher count of yeast was significantly reduced in SCC, with a maximum reduction of 1.5–2.0 log values after processing at 400 and 600 MPa. In general, yeast cells are very sensitive to high-pressure treatment, owing to their morphology, namely, being susceptible to death by high pressure [17]. However, Sohn and Lee [5] indicated that the yeast in kimchi was not affected by HHP treatment (200–600 MPa) and maintained a certain value of count (3 log CFU/g).

### 3.2. Effects of High Pressure on the Appearance and Color Value of Mustard Pickle

The changes of appearance and color values in mustard pickle samples after HHP are presented in Figure 1 and Table 2, respectively. Compared with mustard pickle samples in the control group, high-pressure treatment gave the mustard pickle samples a brighter appearance (Figure 1). Additionally, the *L**, *a**, *b**, and *W* values of the control group were 39.34, 3.74, 30.62, and 33.04, respectively. After HHP treatment, the *L** values of the samples increased with the increasing pressure, reaching a maximum of 46.29 after processing at 600 MPa. The *a** values displayed a tendency to decrease slightly with increasing pressure, reaching a minimum of 2.09 after processing at 600 MPa, while the *b** values tended to decrease slightly with increasing pressure, reaching a minimum of 21.88 after processing at 600 MPa. The *W* values exhibited the same trend as the *L** value, namely, the *W* values increased with increasing pressure, reaching a maximum of 41.95 after processing at 600 MPa. The Δ*E* (color difference) increased from 6.83 after processing at 200 MPa to 45.72 at 600 MPa. The aforementioned findings and their comparison with the changes in appearance presented in Figure 1 revealed that the high-pressure processing made the mustard pickle look brighter and whiter.

However, Park et al. [6] indicated that there were no significant statistical differences in redness and yellowness between HHP-unprocessed and HHP-processed kimchi (100–500 MPa). The *L** values (lightness) of HHP-processed kimchi were significantly lower than those of HHP-unprocessed kimchi, and the appearance was darker. Such findings were different from the present experiment. This may be because many ingredients (e.g., paprika) were added to the kimchi during manufacturing to give it a dark red appearance, which affected or obscured the changes induced by the HHP treatment. The increase in the lightness (*L** value) and whiteness (*W* value) of HHP-processed mustard pickles in this study may be relevant to the increase in water content due to changes in plant tissue structure induced by high pressure, which increased the hydration capacity and allowed external water to be injected into the tissue [18].

### 3.3. Effects of High Pressure on the Texture of the Mustard Pickle

The texture values of mustard pickle samples after HHP processing are shown in Table 3. Regarding the control group, the hardness was 222.75 (g), cohesiveness was 0.48, springiness was 3.70 (mm), and chewiness was 3.87 (mJ). After HHP treatment, the hardness of the samples increased as the pressure increased, reaching a maximum of 618.80 (g) after processing at 600 MPa. However, the cohesiveness and springiness of the samples after HHP processing were not significantly different from those of the control group (*p* > 0.05), namely, HHP had little effect on the cohesiveness and springiness of mustard pickles. However, the chewiness exhibited the same trend as the hardness, namely, the higher the pressure, the higher chewiness, reaching a maximum of 9.94 (mJ) after processing at 600 MPa. In summary, it was revealed that the HHP gave the mustard pickle a hard texture. The results showed that mustard pickle processed by high pressure had a better taste. Sohn and Lee [5] found that the high-pressure processing of kimchi at 400 MPa had no effect on the texture of kimchi, but the cutting force increased after processing at 600 MPa, namely, the texture was harder. However, Park et al. [6] indicated that there existed no statistically significant difference in hardness between HHP-unprocessed and HHP-processed kimchi (100–500 MPa). Such findings differed from the present experiment. This may be attributed to the vegetable ingredients of kimchi, which are different than the mustard pickle in this present study, and because multiple ingredients added during manufacturing may have affected or interfered with the effects of the HHP processing. In addition, the increase in the hardness values of the samples may be relevant to the change in plant cell morphology during the pressurization process [5]. Another explanation is that the high pressure induced the release of pectin methylesterase from the tissue to catalyze the demethylation of the highly methylated pectin. The resulting product, de-esterified pectin (low-methylated pectin), can form a gel network with divalent ions [18,19]. Overall, the reason why the high pressure increases the hardness of the sample is that the pressure promotes the rearrangement of the plant cell structure and releases pectin methylesterase activity to produce de-esterified pectin products, which then form a stable network structure with divalent metal ions.

### 3.4. Quality Changes of HHP-Processed Mustard Pickle Samples after Storage at 25 °C

#### 3.4.1. Microbial Quality

The variations of APC after HHP processing (400 and 600 MPa) of mustard pickles stored at 25 °C for 60 days are presented in Figure 2a. The initial viable bacteria count of the control group was about 6.8 log CFU/g, which increased slowly as the storage time increased and reached about 7.7 log CFU/g after 60 days of storage. In contrast, the groups processed at 400 and 600 MPa exhibited the same APC growth trend: both increasing rapidly from day 0 to day 6, at which time the APC levels were similar to those of the control group, and then increasing slowly with the increase in storage time. Furthermore, no significant differences with the control group were observed throughout the storage period (*p* > 0.05). The results of this present study are similar to those of previous studies on HHP-processed SCC [7], namely, the groups processed at 400 MPa exhibited a rapid increase in the count of organisms at the beginning of storage at 27 °C, and the findings were similar to those of the HHP-unprocessed group after about 15 days.

The variations of LAB values for HHP-processed (400 and 600 MPa) mustard pickles stored at 25 °C for 60 days are presented in Figure 2b. The growth trend of LAB was similar to that of APC, namely, the LAB of the control group increased slowly with the increasing storage time. LAB in 400 MPa and 600 MPa HHP processing groups exhibited the same growth trend: a rapid increase from day 0 to day 6, at which time the LAB counts were similar to those of the control group and then increased slowly with increasing storage time. Furthermore, they exhibited no significant differences from the control group throughout the storage period (*p* > 0.05). The results of our study were similar to those of previous studies on HHP-treated SCC [7], namely, the lactic acid bacteria count of the 400 MPa HHP-treated (10 min) samples increased rapidly at the beginning of storage at 27 °C and reached approximately the same level as the HHP-unprocessed group after about 6 days. In conclusion, the rapid increase in the number of bacteria in the HHP-processed samples at the beginning of the storage period may be because some of the sub-lethally injured microorganisms repaired the damage caused by HHP processing and overgrew during the storage period [20]. Therefore, it was concluded that the increase in the number of APC and LAB bacteria in the HHP samples was due to the regeneration of the residual and injured bacterial cells.

The variations in the yeast count in mustard pickles stored at 25 °C for 60 days induced by HHP processing (400 and 600 MPa) are presented in Figure 2c. In the control group, yeast counts increased gradually as the storage time increased, reaching about 5.5 log CFU/g after 60 days of storage. However, the yeast did not grow in the 400 and 600 MPa high-pressure groups until 24 days. Moreover, yeast counts were detected on days 30 and 42, respectively, and then increased slowly to reach the final levels (counts below 2.0 log CFU/g). The high-pressure treatment significantly slowed the growth of yeast in the mustard pickle when stored at 25 °C. However, Li et al. [7] treated SCC with HPP processing and found that yeast counts in the control group decreased and then increased when stored at 27 °C for 90 days. No growth was observed in the high-pressure group (400 MPa) during the storage period, which the authors suggested could be attributed to the yeast in SCC, which may be sensitive to high pressure. In general, yeast does not cause food poisoning; however, it plays a vital role in food spoilage, notably in acidic foods [21]. In contrast, yeast can produce gas in SCC and cause spoilage by swelling [7]. This study revealed that the HHP processing obviously retarded the growth of yeast in mustard pickles when stored at 25 °C, which in turn delayed the deterioration of mustard pickles.

#### 3.4.2. Chemical Quality

The variation of pH and TVBN values in mustard pickles stored at 25 °C induced by HHP treatment (400 and 600 MPa) is illustrated in Figure 3. The pH value of initial samples was about 4.9, and the pH values in the control group increased slowly with time, reaching about 5.92 at the 60 days of storage. However, the pH values of the 400 and 600 MPa high-pressure groups did not change obviously during storage and were significantly lower than those of the control group during the same storage time (Figure 3a). The findings indicated that the high-pressure treatment significantly inhibited the pH increase in mustard pickles during storage at 25 °C. The possible reason was that HHP treatment inhibited the growth of yeast (Figure 2c), especially some yeasts that produced biogenic amines that are rich in mustard pickle, thus delaying the rise of the pH in HHP samples [2].

In terms of TVBN values, the initial TVBN value of the mustard pickle samples was about 20–25 mg/100 g. The TVBN values of the control group continued to increase with time and reached about 56 mg/100 g after 60 days of storage. However, the TVBN values of the 400 and 600 MPa high-pressure groups increased slowly during storage and reached about 42 mg/100 g and 33 mg/100 g after 60 days of storage, respectively. Moreover, they were significantly lower than those in the control group during the same storage time (Figure 3b). The findings revealed that HHP treatment significantly inhibited the increase in TVBN values in mustard pickles during storage at 25 °C. Kung et al. [2] found biogenic-amine-producing yeast (*Candida* spp.) in mustard pickle samples in Taiwan. On the basis of Figure 2b, there was no significant difference in LAB counts between the control group and the 400 and 600 MPa HHP groups during storage, indicating that LAB had little effect on the pH value of the samples. However, considering the findings depicted in Figure 2c, it was speculated that the increase in pH and TVBN values in the control group may have been due to the growth and amine production of biogenic-amine-producing yeast in mustard pickles during storage.

### 3.5. HTS Analysis of High-Pressure Samples

Figure 4 presents the results of species annotation of mustard pickle samples in the control 400 MPa and 600 MPa high-pressure groups after 60 days of storage at 25 °C. In terms of phylum level, the dominant bacteria in the D0-control, D60-control, 400 MPa (D60-400 MPa), and 600 MPa (D60-600 MPa) groups were *Firmicutes*, which accounted for 97%, 100%, 100%, and 94%, respectively (Figure 4a). In terms of class level, the dominant bacteria in the D0-control, D60-control, D60-400 MPa, and D60-600 MPa groups were *Bacilli*, with proportions of 97%, 100%, 100%, and 94%, respectively (Figure 4b). In terms of order level, the dominant bacteria in the D0-control, D60-control, D60-400 MPa, and D60-600 MPa groups were *Lactobacillales*, a proportions of 97%, 96%, 100%, and 94%, respectively (Figure 4c). In terms of family level, the dominant bacteria in the D0-control, D60-control, D60-400 MPa, and D60-600 MPa groups were *Lactobacillaceae*, with proportions of 97%, 96%, 100%, and 94%, respectively (Figure 4d). In terms of genus level, the dominant bacteria in the D0-control group were *Lactiplantibacillus* (74%), *Latilactobacillus* (12%), and *Levilactobacillus* (6%). The dominant bacteria were *Companilactobacillus* (80%) and *Lactiplantibacillus* (9%) in the D60-control group, *Lactiplantibacillus* (92%) and *Levilactobacillus* (8%) in the D60-400 MPa group. Moreover, *Levilactobacillus* (52%), *Pediococcus* (17%), and *Lactiplantibacillus* (17%) were the dominant bacteria in the D60-600 MPa group (Figure 4e).

Figure 5 presents the results of species annotation for mustard pickle samples in the control 400 MPa and 600 MPa groups after 60 days of storage at 25 °C. The OTU data of the first 35 bacterial genera were selected to create a heat map. The heat map is presented as two-dimensional data, where each column represents an experimental group, and each row represents an OTU. The main genera in the D0-control group were *Klebsiella*, *Pectobacterium*, *Raoultella*, *Latilactobacillus*, *Neochroococcus,* and *Weissella*. The proportions dropped sharply after 60 days of storage. Eventually, *Companilactobacillus*, *Citrobacter*, *Secundilactobacillus*, *Cytobacillus*, and *Lentilactobacillus* were dominant in the D60-control group. In addition, a significant difference existed in the bacterial community structure between the D60-400 MPa group and the 600 MPa group. *Lactiplantibacillus* predominated in the D60-400 MPa group. However, *Levilactobacillus*, *Sphingomonas*, *Paracoccus*, *Methylorubrum*, *Diaphorobacter*, *Staphylococcus*, *Cloacibacterium*, *Sphingobacterium*, *Enterobacter*, *Leuconostoc*, and *Pediococcus* predominated the D60-600 MPa group. In general, while the bacterial composition varied across mustard pickle samples in groups processed at different high pressures, *Lactobacillaceae* predominated in all groups. Chao et al. [22] pointed out that *Lactobacillus*, *Pediococcus*, *Weissella*, and *Leuconostoc* are the major microbiota in Taiwanese mustard pickles during fermentation. The three dominant bacterial genera in kimchi were reported to be *Leuconostoc*, *Lactobacillus*, and *Weissella* [23,24]. Moreover, researchers analyzed Sichuan paocai brine and Chongqing radish paocai brine using PacBio SMRT and found *Lactobacillus* to be the primary predominant bacterium [12,25]. The findings of this present study indicated that HHP altered the bacterial flora of *Lactobacillus* in mustard pickle samples during storage. To the best of our knowledge, this is the first research to elucidate the microbiota of mustard pickles processed at different high pressures during storage.

## 4. Conclusions

The results of this research showed that the counts of APC and LAB in the high-pressure treatment samples decreased significantly with increasing pressure, while the count of yeast decreased to undetectable levels. Moreover, the texture was hard, and the color and appearance were brighter and whiter. Furthermore, the APC and LAB rapidly recovered growth during the storage period of the HHP-processed mustard pickle, whereas the yeast growth and the increase in pH and TVBN values were significantly inhibited. The result indicated that the main effect of the HHP applied to mustard pickle is whitening. Moreover, the HTS analysis revealed that the HHP processing changed the bacterial flora of lactic acid bacteria during the storage period. In conclusion, the application of the HHP technique to mustard pickle products can effectively improve the texture and color, retard the growth of yeast, and delay the deterioration of quality during storage. Therefore, it has the potential to be used as an alternative treatment technique to sulfite addition in mustard pickles.

## Figures and Tables

**Figure 1 foods-12-00317-f001:**
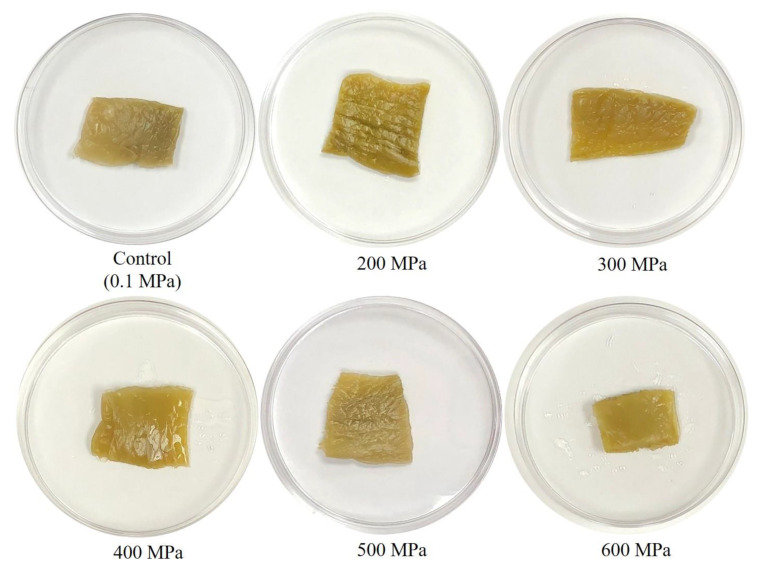
Changes of the appearance in fresh mustard pickles after HHP processing at 200, 300, 400, 500, and 600 MPa for 5 min.

**Figure 2 foods-12-00317-f002:**
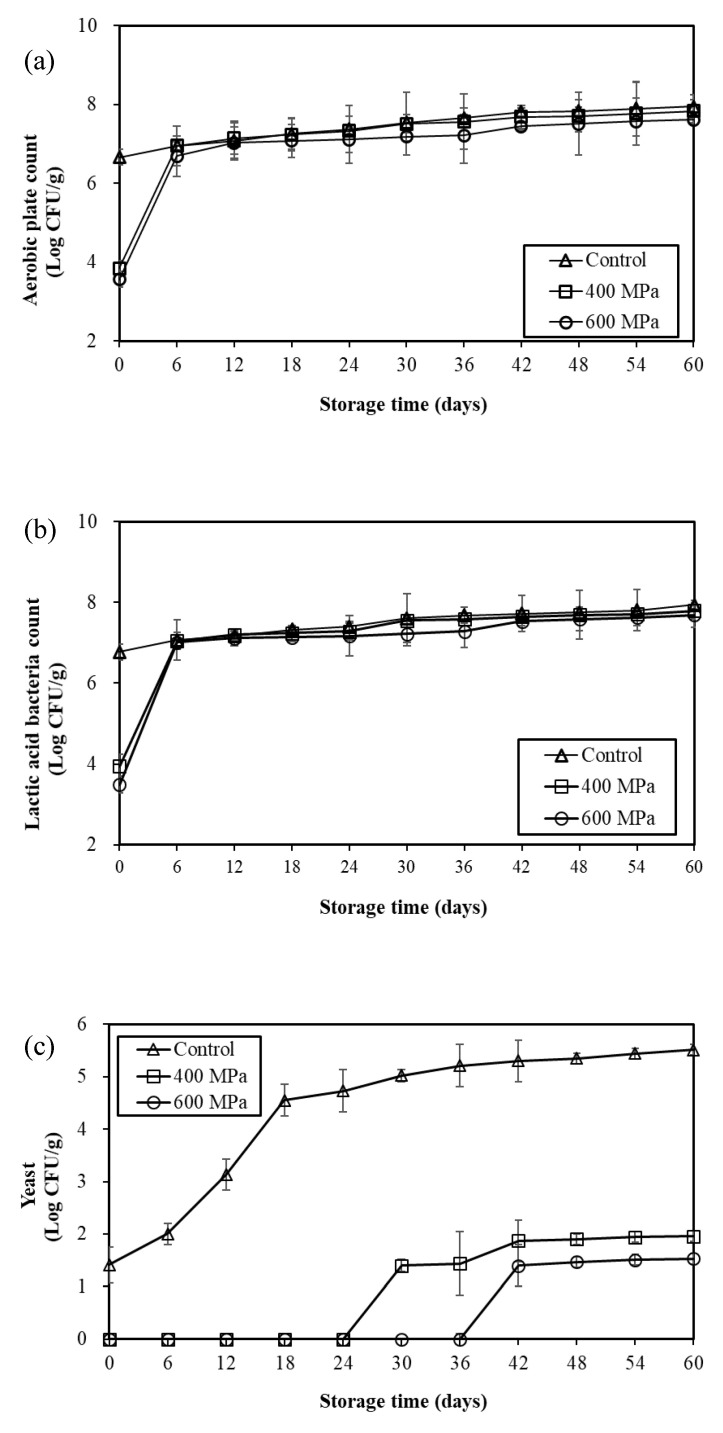
Changes of (**a**) aerobic plate count (APC), (**b**) lactic acid bacteria count, and (**c**) yeast in mustard pickles treated with HHP at 400 and 600 MPa for 5 min during storage at 25 °C.

**Figure 3 foods-12-00317-f003:**
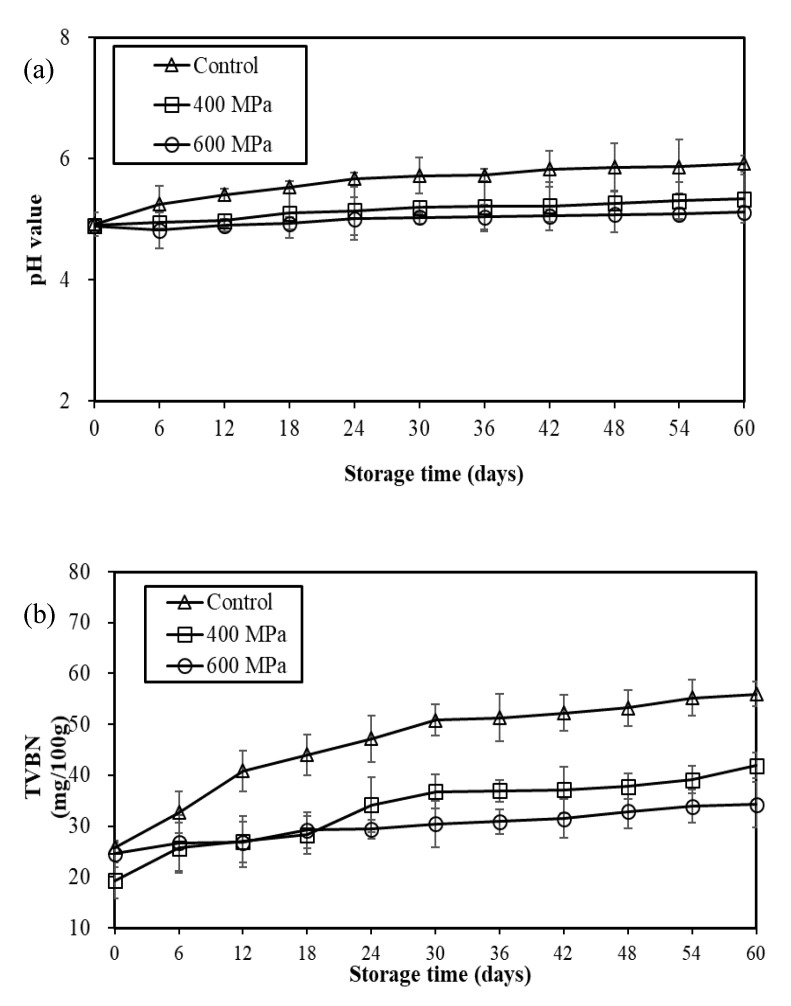
Changes of (**a**) pH value and (**b**) total volatile basic nitrogen (TVBN) in mustard pickles treated with HHP at 400 and 600 MPa for 5 min during storage at 25 °C.

**Figure 4 foods-12-00317-f004:**
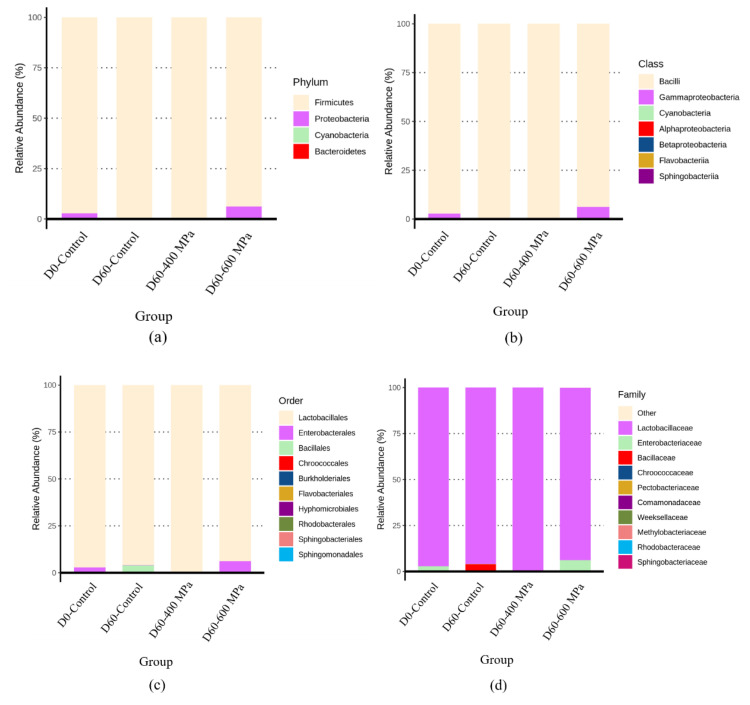
Bacteria species annotation of mustard pickle products on the levels of (**a**) phylum, (**b**) class, (**c**) order, (**d**) family, and (**e**) genus, as a result of the non-HHP group (D0-Control), non-HHP group stored at 25 °C for 60 days (D60-Control), and 400 MPa group (D60-400 MPa) and 600 MPa group (D60-600 MPa) stored at 25 °C for 60 days.

**Figure 5 foods-12-00317-f005:**
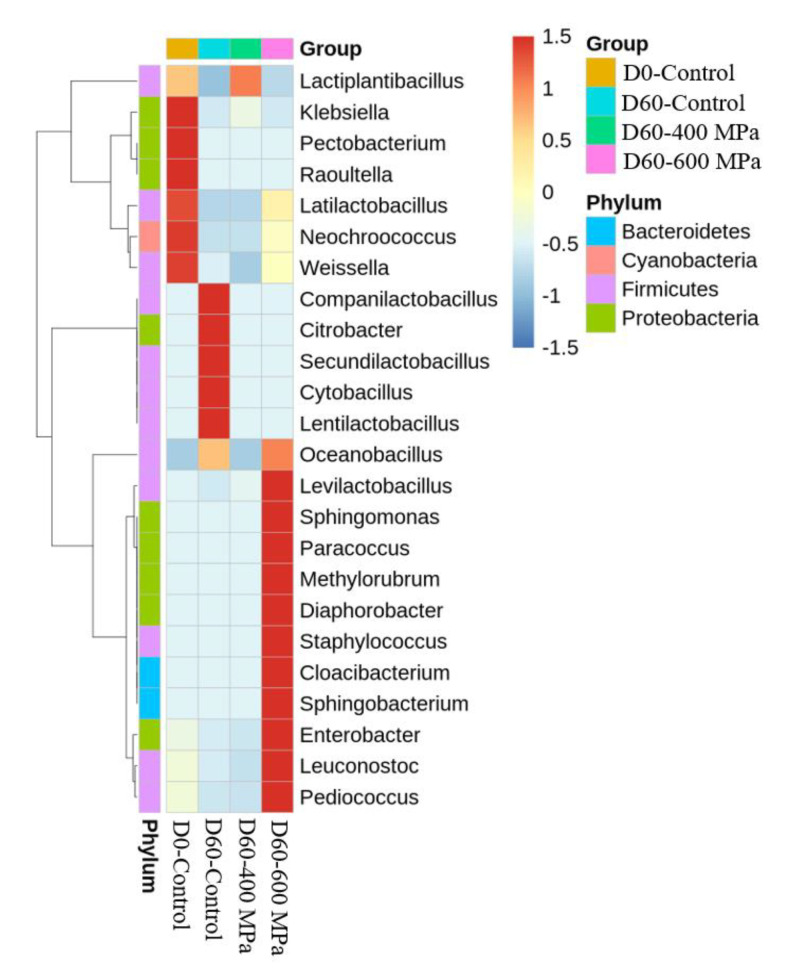
Heat map of microorganism species abundance on the genus level, as a result of the non-HHP group (D0-control), non-HHP group stored at 25 °C for 60 days (D60-control), and 400 MPa group (D60-400 MPa) and 600 MPa group (D60-600 MPa) stored at 25 °C for 60 days.

**Table 1 foods-12-00317-t001:** Value of aerobic plate count (APC), lactic acid bacteria count (LAB), and yeast in fresh mustard pickles after HHP processing at 200, 300, 400, 500, and 600 MPa for 5 min.

Treatments	APC (log CFU/g)	LAB (log CFU/g)	Yeast (log CFU/g)
Control (0.1 MPa)	6.99 ± 0.02 ^a,^*	6.99 ± 0.08 ^a^	1.41 ± 0.34
200 MPa	6.73 ± 0.06 ^b^	6.56 ± 0.02 ^b^	<1
300 MPa	6.32 ± 0.01 ^c^	6.53 ± 0.12 ^b^	<1
400 MPa	4.23 ± 0.10 ^d^	4.67 ± 0.06 ^c^	<1
500 MPa	4.03 ± 0.19 ^e^	3.97 ± 0.06 ^d^	<1
600 MPa	3.81 ± 0.01 ^f^	3.75 ± 0.08 ^e^	<1

* Each data point is the average ± SD (three samples). Within the same column, different letters present significant differences (*p* < 0.05). APC: aerobic plate count; LAB: lactic acid bacteria count.

**Table 2 foods-12-00317-t002:** Values of *L**, *a**, *b**, *W*, and Δ*E* in fresh mustard pickles after HHP processing at 200, 300, 400, 500, and 600 MPa for 5 min.

Treatments	*L**	*a**	*b**	*W*	Δ*E*
Control (0.1 MPa)	39.34 ± 0.12 *^,d^	3.74 ± 0.62 ^a^	30.62 ± 1.61 ^a^	33.04 ± 0.19 ^e^	-
200 MPa	42.49 ± 0.08 ^c^	2.96 ± 0.06 ^a^	28.19 ± 0.18 ^a^	35.16 ± 0.53 ^d^	6.83 ± 2.02 ^e^
300 MPa	42.90 ± 0.05 ^c^	3.12 ± 0.83 ^a^	29.76 ± 1.11 ^a^	35.27 ± 0.67 ^d^	10.58 ± 4.68 ^d^
400 MPa	43.57 ± 0.07 ^b^	2.33 ± 0.11 ^b^	26.33 ± 0.03 ^b^	38.37 ± 0.04 ^c^	15.41 ± 1.17 ^c^
500 MPa	46.08 ± 0.09 ^a^	2.29 ± 0.03 ^b^	24.66 ± 0.08 ^b^	39.95 ± 0.07 ^b^	24.70 ± 1.73 ^b^
600 MPa	46.29 ± 0.07 ^a^	2.09 ± 0.07 ^b^	21.88 ± 1.83 ^c^	41.95 ± 0.70 ^a^	45.72 ± 7.89 ^a^

* Each data point is the average ± SD (three samples). Within the same column, different letters present significant differences (*p* < 0.05). *L**: lightness; *a**: redness; *b**: yellowness; *W*: whiteness; Δ*E*: total color difference.

**Table 3 foods-12-00317-t003:** Values of hardness, cohesiveness, springiness, and chewiness in fresh mustard pickles after HHP processing at 200, 300, 400, 500, and 600 MPa for 5 min.

Treatments	Hardness (g)	Cohesiveness	Springiness (mm)	Chewiness (mJ)
Control (0.1 MPa)	222.75 ± 52.56 *^,d^	0.48 ± 0.05 ^a^	3.70 ± 0.12 ^a^	3.87 ± 0.08 ^c^
200 MPa	334.33 ± 48.02 ^c^	0.52 ± 0.07 ^a^	3.84 ± 0.08 ^a^	6.52 ± 1.15 ^b^
300 MPa	465.08 ± 29.56 ^b^	0.44 ± 0.06 ^a^	3.82 ± 0.08 ^a^	7.50 ± 0.77 ^b^
400 MPa	471.00 ± 34.15 ^b^	0.51 ± 0.11 ^a^	3.90 ± 0.01 ^a^	9.08 ± 1.84 ^a^
500 MPa	604.17 ± 82.46 ^a^	0.38 ± 0.03 ^a^	3.76 ± 0.19 ^a^	8.51 ± 1.49 ^a^
600 MPa	618.80 ± 85.44 ^a^	0.44 ± 0.03 ^a^	3.73 ± 0.13 ^a^	9.94 ± 0.78 ^a^

* Each data point is the average ± SD (three samples). Within the same column, different letters present significant differences (*p* < 0.05).

## Data Availability

The data presented in this study are available on request from the corresponding author. The data are not publicly available due to privacy and ethical reasons.

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
