# Peer review of "Replacing the Addition of Sulfite in Mustard Pickle Products by High-Hydrostatic-Pressure Processing to Delay Quality Deterioration during Storage"

_foods, 2023, doi:10.3390/foods12020317_

Round 1

Reviewer 1 Report

The aim of the study was to assess the use of the high-hydrostatic-pressure (HHP) for pasteurizing and preserving mustard pickle products as an alternative to the conventional method of sulfite addition.

I think the main reason is that sulfites are dangerous allergens; authors should explain this in the manuscript;

Line 43-45: The main manufacturing process includes placing the mustard pickle in a vat, then sprinkling a layer of salt equivalent to approximately 14% of the weight of the mustard….

Does this high amount of salt have a preservative effect? what is the product's aw?

Line 47-48: After fermentation, the outermost leaves of the mustard are removed, and the stems are soaked in sulfite solution for bleaching… 

Is the main purpose of using sulfites for whitening or antimicrobial action? the authors should make this clear;

Line 309-310:  In summary, it was revealed that the HHP gave the mustard pickle a hard texture.

Is this a good or bad thing?

Line 348: in Figure 2 (a, b) APC and LAB counts are substantially similar to the control, this shows that the main effect of the HHP applied to mustard pickle products is whitening; for this reason the authors should also review the conclusions 

Line 388- 391: However, the pH values of the 400 and 600  MPa high-pressure groups did not change obviously during storage and were significantly lower than those of the control group during the same storage time (Fig. 3(a)). The findings indicated that the high-pressure treatment significantly inhibited the pH increase of mustard pickles during storage at 25 °C.

The authors should explain why the high-pressure treatment "inhibit" pH;

Line 409-411: it was speculated that the increase in pH and TVBN values in the control group may be due to the growth and amine production of biogenic amine-producing yeast in mustard pickles during storage.

is only a hypothesis, the authors should at least correlate the pH and TVBN values with the development of yeasts and LAB in their experiment;

Reviewer 2 Report

1. The English quality should be fine-tuned

2. The flow of the abstract should be considered

3. Line No 60 to 105 looking like a discussion, they can be trimmed

4. Better to end the Introduction with the objective of the study

5. Provide the information on the mustard pickle in section 2.1, Like compositions, days after fermentation, and variety of the mustered. This information may be useful for the future studies

6. Give specifications of the vacuum bags used in the study (section 2.1)

7. Tables have to revise, the abbreviations given in the tables should give full names in the footnote 

8. In table 1 titles the values are of which day storage sample? or fresh?

9. Figure 1 is for fresh samples or stored samples?

10. Mention p-values in the figures

11. Try to discuss the reasons for the increased hardness in relation to the application of pressure 

12. Figure 4, font sizes are very small

13. Fine tune conclusion  in relation to your objectives 

Reviewer 3 Report

Change the title. It should be clear to indicate the sulfite replacement.

Give text on effect of HHP on bacterial spores present in samples?

Line 123: provide GPS location for pickle sample collection or production.

Fig 1: increase the resolution.

Fig 2: provide standard deviation for the line graphs. Do it for all.

Round 2

Reviewer 1 Report

I thank the authors for their revisions; I think that in the mustard pickle after the sulphites the risk of biogenic amines remains